# Analysis of Land Use Change and Its Impact on the Hydrology of Kakia and Esamburmbur Sub-Watersheds of Narok County, Kenya

**Nzitonda Marie Mireille [1,*], Hosea M. Mwangi [2], John K. Mwangi [3] and John Mwangi Gathenya [2]**

[1] Civil Engineering Department, Pan African University–Institute for Basic Sciences Technology and Innovation/Jomo Kenyatta University of Agriculture and Technology, P.O. Box 62000-00200 Nairobi, Kenya

[2] Soil, Water & Environmental Engineering Department/Jomo Kenyatta University of Agriculture and Technology, P.O. Box 62000-00200 Nairobi, Kenya; hmwangi@jkuat.ac.ke (H.M.M.); j.m.gathenya@jkuat.ac.ke (J.M.G.)

[3] Department of Civil, Construction and Environmental Engineering/Jomo Kenyatta University of Agriculture and Technology, P. O. Box 62000-00200 Nairobi, Kenya; joymwa86@yahoo.com

[*] Correspondence: nzitondam@yahoo.fr; Tel.: +254-743197758 or +257-79363832

**Abstract:** Narok town is one of the places in Kenya which experience catastrophic floods. Many lives have been lost and valuable property destroyed in recent years. Change in land use/land cover upstream of the town area may have contributed significantly to the severity and frequency of flooding events. Runoff, which contributes to floods in Narok town, comes from Kakia and Esamburmbur sub-catchments of Enkare Narok watershed. The objective of this study was to assess the impact of land use/land cover change on the hydrology of Kakia and Esamburmbur sub-watersheds. To detect land use/land cover change, Landsat satellite images from 1985 to 2019 were used. Using supervised classification in Erdas Imagine 2014, land use of the study area was classified into four classes, i.e., forest, rangeland, agriculture and built-up areas. Five land use maps (1985, 1995, 2000, 2010, and 2019) were developed and used to perform land use change analysis. There was rampart conversion of forest to other land uses. Between 1985 and 2019, the forest and rangeland declined by 40.3% and 25.6% of the study area, respectively, while agriculture and built-up areas increased by 55.2% and 10.6% of the study area respectively. Analysis of soil hydrological properties indicate that the infiltration rate and soil hydraulic conductivity were greatest in forest than in other land use types. The basic infiltration rate in forest land was 89.1 cm/h while in rangeland and agricultural land, it was 7.9 cm/h and 15 cm/h respectively. At the top-soil layer, average soil hydraulic conductivity under forest was 46.3 cm/h, under rangeland, 2.6 cm/h and under agriculture, 4.9 cm/h. The low hydraulic conductivity in rangeland and agriculture was attributed to compaction by farm machinery (tractors) and livestock respectively. An interesting observation was made in rangelands where the top layer (0–20 cm) had a higher bulk density and a lower hydraulic conductivity as compared to the next deeper layer (20–40 cm). This was attributed to the combined impact of compaction and localised pressure by hooves of livestock which only have an impact on the top layer. The findings of this study show that land use has a major impact on soil hydrological properties and imply that the observed land use changes negatively affected the soil hydrological properties of the watershed. The decreased infiltration in the increasing areas of degraded land (mainly agriculture and rangeland) and increase in built-up area in Narok town are the possible causes of the increased flood risk in Narok town. It is recommended that flood risk management strategies in Narok town include watershed management to enhance water infiltration.

**Keywords:** land use/cover change; soil hydrological properties; watershed; flood management

## 1. Introduction

The negative impacts of land use/land cover change on the environment is a major challenge that communities all over the world have to deal with. Population growth is a major driver of land use/land cover change. Population growth commonly leads to urbanization and expansion of agricultural land with the aim of increasing food production [1]. Another driver of land use/land cover change is industrialization, which increases pressure on resources and change in socio-economic organization [2]. Land use/land cover change may negatively impact some natural resources (e.g., water and soils) which may eventually lead to soil degradation and ultimately, reduction in food production [3].

Land use/land cover change may negatively affect the environment and livelihoods. One of the problems caused by land use/land cover change is soil degradation. In many cases, land degradation starts when a particular original land cover (e.g., forest) is converted to other land uses such as farmlands, grazing lands, settlement and urban expansion [4]. When the forest is converted to other land uses, such as farmland and wood land, there is a significant loss of nutrients and biodiversity, and soil erosion is increased, which lead to soil degradation [5].

The extent of soil degradation depends on the intensity, pattern and type of land use/land cover change. For example, conversion of forest to cultivated land can cause a reduction or depletion of soil nutrients such as organic matter, total nitrogen and phosphorus [5]. The conversion of forest to cultivated land or grazing may decrease the infiltration capacity and soil moisture content because of change in soil structure caused by surface soil compaction. A study done by [6] in Ethiopia shows that infiltration in cultivated land and pasture was 70% lower compared to the forest and moisture content in these two land uses (cultivated land and pasture) was 45% lower compared to the forest. The bulk density in cultivated land and pasture was 13–20% greater compared to the forest. Other studies have reported an increase of bulk density in cultivated land compared to fallow land and woodland and soil organic matter being greater in fallow land and woodland than in cultivated land [7]. Land use/land cover change may also have an effect on soil hydraulic conductivity. For example, an analysis of soil samples taken in different land uses (natural forest, corn, grassland, hazelnut garden) in the Dagdami river catchment in Turkey showed that the saturated hydraulic conductivity in natural forest was 82.4 cm/h compared to that of grassland (8.4 cm/h), hazelnut garden (11.5 cm/h) and corn field (30.0 cm/h) [8]. In another study [9], the soil hydrological properties of three land uses (degraded grass covered field, traditional coffee plantation and primary forest) were studied in the Talgua River watershed in Honduras and the results showed that hydraulic conductivity was lower in degraded grass compared to non-degraded lands. Under a traditional coffee plantation and primary forest, the soils had high infiltration capacities and readily conducted water vertically at rates of 109 and 840 mm/h, while in land under degraded grass, the hydraulic conductivity was between 8 and 11mm/h [9].

Land use/land cover change has a significant effect on the hydrological behaviour in a catchment in terms of runoff generation and soil erosion caused by runoff. A decrease in soil infiltration capacity leads to increased runoff and soil erosion. The more the runoff the more the erosive power and the more the soil erosion. In term of soil loss, the transition of other land use/land cover to cropland increases the soil loss, while forest is a good land use for soil conservation [10]. A study done in the high Côa river catchment of Portugal reported that there was a significant amount of runoff and soil erosion in arable land and coniferous afforestation compared to fallow land, shrub cover, recovering autochthonous vegetation, and pastureland [11]. In another study done by [12] in the Finchaa River basin located in western part of Oromia state, Ethiopia, it was found that afforestation reduced soil erosion, sediment yield and runoff generation. The average sediment yield and surface runoff in afforestation were 56.041 tons/ha and 227.89 mm, respectively, compared to the conversion of land use to agriculture (105.243 tons/ha and 366.14 mm, respectively). A review of several studies in East Africa by [13] found that forest cover loss increased annual discharges and surface runoff by 16 ± 5.5% and 45 ± 14%, respectively. Another study done in the Nyando River in Lake Victoria drainage basin in Kenya showed a significant increase in peak discharge in an area under deforestation located in the upstream area [14].

Land use/land cover change can also influence the occurrence of floods. A decrease in forest areas and an increase in urban built-up areas contribute to increased flood events [15]. In recent years, there has been many events of flooding in Narok which have caused the loss of many lives and massive damage to property, especially in Narok town. For example, in March 2013, 15 lives were lost and about 350 people were displaced by flooding [16]. Another flooding event that Narok town experienced was on 28 April 2015, when 15 people were recorded dead [17]. The increase in the magnitude and frequency in flooding events may have been caused by change in land use/land cover in the upstream area. However, to the best of our knowledge, no study has investigated the extent of land degradation in the catchment as possible cause of flooding in Narok town. This is a gap in knowledge that this study aims to fill. Therefore, as a step towards developing measures for mitigating damages caused by flood, the aim of this study was to investigate how land use/land cover has changed in Kakia and Esamburmbur sub-catchments in Narok over 34 years, which may be the cause of flooding in Narok town. The study also investigated how the impact of different land uses on soil hydrological properties (i.e., hydraulic conductivity, infiltration, bulk density) as an indicator of how land use change over the years has affected the hydrology of the watershed.

## 2. Materials and Methods

### 2.1. Study Area

The study area is located in Narok County in south-west Kenya and it is composed of two sub-catchments, namely Kakia (30.5 km$^2$) and Esamburmbur (15.7 km$^2$) (Figure 1). The seasonal Kakia and Esamburmbur streams flow through Narok town and meet a few meters before draining into Enkare Narok River (Figure 1). Enkare Narok River is a permanent river that flows from the Mau forest and through Narok town where the two seasonal tributaries (Kakia and Esamburmbur) join. The seasonal streams run through the main shopping Centre and act as storm drain channels for the town before emptying into Narok River. The drainage area that contributes to flooding in Narok town is comprised of these two sub catchments.

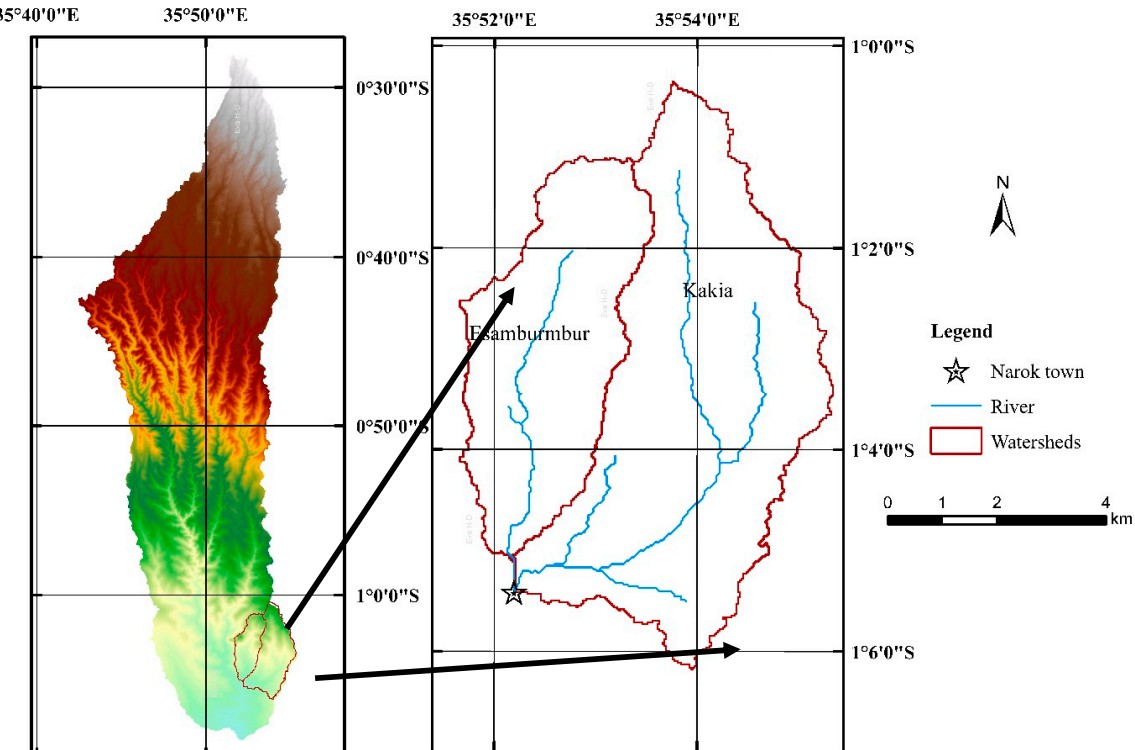

**Figure 1.** Kakia and Esamburmbur sub-catchments.

*2.2. Data Preparation*

Landsat 5 and 8 at a resolution of 30 m were used for land use land cover classification. Five sets of Landsat images from 1985, 1995, 2000, 2010, and 2019 were obtained from United States Geological Survey (USGS) and used to detect land uses/land cover change in Kakia and Esamburmbur sub-catchments. The images were downloaded from earth explorer site (http://earthexplorer.usgs.gov/).

The downloaded images (Table 1) were taken during the dry season period (January–mid-March) in order to avoid difficulties that may arise in differentiating some vegetation-based land use classes (e.g., forest and crops) for images taken during rainy seasons.

**Table 1.** Details of acquired satellite images.

| Satellite id | Sensor id | Path/Row | Acquisition Date | Spatial Resolution |
| --- | --- | --- | --- | --- |
| Landsat 5 | TM | 169/61 | 9 January 1985 | 30 |
| Landsat 5 | TM | 169/61 | 6 February 1995 | 30 |
| Landsat 7 | ETM | 169/61 | 12 February 2000 | 30 |
| Landsat 5 | TM | 169/61 | 30 January 2010 | 30 |
| Landsat 8 | OLI/TIRS | 169/61 | 12 March 2019 | 30 |

In change detection, the thematic accuracy of the output is directly proportional to the product of the categorical accuracies and relative spatial accuracy of the input classified images. Classification often depends on absolutely accurate image datasets in, for example, the combinination of the imagery with ancillary large-scale maps to refine the classifications and/or for an accuracy assessment [18].

In order to correct random and residual errors resulting in a map-accurate dataset, Orthocorrection of TM/ETM imagery was conducted within ERDAS Imagine using the Landsat model within the Geometric Correction facility. Georeferencing, mosaic, and layer stack were done using Erdas image 2014 software (Hexagon Geospatial, Madson, AL, USA) (Figure 2). As our study area is a small part of the image, a subset (Figure 2) was done in order to remain within the area of interest.

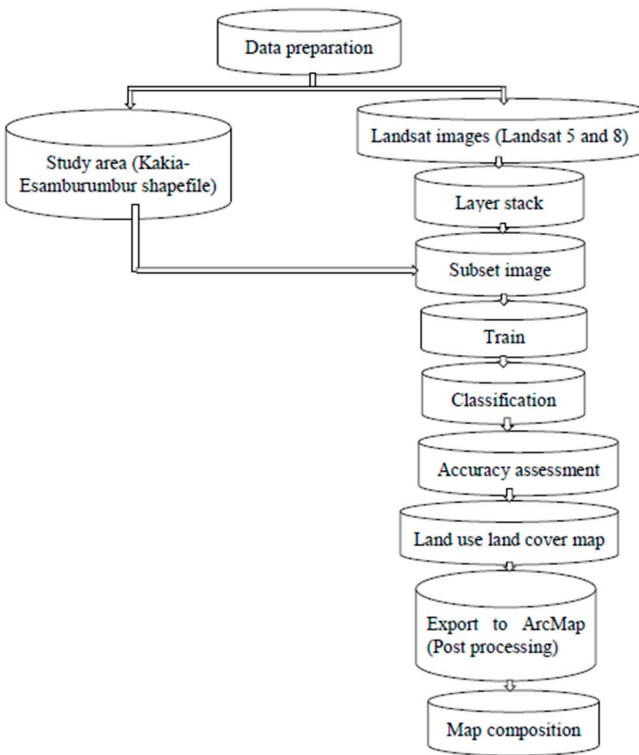

**Figure 2.** Data analysis flowchart.

### 2.3. Land Use Land Cover Classification

After finishing layer stack and subset, the study area map was imported in Erdas imagine software 2014 for land use land cover classification (Figure 2). A supervised classification method was used to classify the land use/cover maps for the land use land cover change analysis in Kakia and Esamburmbur sub-catchments. Supervised classification has been found to be more accurate than the unsupervised classification method [19]. In supervised classification, useful information categories are distinct first, and then their spectral separability is examined while in the unsupervised approach, the computer determines a spectrally separable class, and then defines its information value [20].

The relatively small size of the study area and the authors' knowledge of the catchments were also considered in the selection of the supervised classification method. The Maximum Likelihood algorithm [20] was used for supervised classification. In this method, the probability of each pixel belonging to a class is calculated and the values of the pixels compared.

A pixel is assigned to a class where the probability value is highest. In this method, it is assumed that all the input bands have a normal distribution and it is highly efficient when it comes to classifying the satellite image, particularly the multi-spectral images [21]. A signature editor was created and many training areas were used to classify different classes present on the map. Band combinations were helpful to differentiate between land use classes.

The study area was classified into four land uses classes (forest, rangeland, agriculture and built-up (Table 2).

**Table 2.** Land use/cover scheme.

| Land Use/Cover Type | Description |
| --- | --- |
| Built-up areas | Land covered by buildings (residential, commercial), industrial area, road and other urbans |
| Rangeland | Land covered by grasslands, shrub lands, woodlands, wetlands, and that are grazed by domestic livestock or animals |
| Forest | Lands dominated by trees with a percent cover >60% and height exceeding 2 m |
| Agriculture | Lands covered with temporary crops followed by harvest period, Crop fields and pastures |

### 2.4. Accuracy Assessment

The accuracy assessment was an important step because it confirmed the results found after land use land cover classification. Before using a map, its evaluation must be done to determine its accuracy and how accurate it should be to sufficiently meet the requirements for the intended application [22]. A study conducted by the Food and Agriculture Organisation of the United Nations (FAO) on map accuracy assessment and area estimation showed three measures which are judged important. Those measures are overall accuracy, which is the proportion of area classified correctly, and thus refers to the probability that a randomly selected location on the map is classified correctly (Equation (1)), and user's accuracy, which is the proportion of the area classified as class $i$ that is also class $i$ in the reference data (Equation (2)). It provides users with the probability that a particular area of the map of class $i$ is also that class on the ground. The last one is producer's accuracy, which is the proportion of area that is reference class $j$ and is also class $j$ in the map (Equation (3)). It is the probability that class $j$ on the ground is mapped as the same class [23].

$$A = \sum_{j=1}^{q} p_{jj} \tag{1}$$

$$U_i = p_{ii}/p_i \tag{2}$$

$$P_j = p_{jj}/p_j \tag{3}$$

where A is the overall accuracy; $U_i$ is the user's accuracy; $P_j$ is the producer's accuracy; $p_{ii}$ is the number of class *i* classified on the map and also on the ground; $p_{jj}$ is the number of class j on the ground and also classified as the same class on the map; $p_i$ is the classified total of class *i* on the map and $p_j$ is the total number of class j on the ground.

In order to detect if the land use/cover classification maps were going to be useful, an accuracy assessment was done in ERDAS software. Both outfield and infield accuracy assessments were used. For four past images (1985, 1995, 2000, and 2010), an outfield assessment was done by comparing the reference image with the classified image with 100 stratified random points based on ground truth data. For the current image, infield accuracy was done with 50 points. The coordinates of those points were taken within the study area and in known land uses and they were then added in the classified image (2019) in order to check its accuracy. Four accuracy measures (Overall accuracy, User accuracy, Producer accuracy and Kappa) were quantified by using an Error matrix.

## 2.5. Impact of Land Use/Cover on Soil Hydrological Properties

An analysis was carried out to investigate the impact of land use on soil hydrological properties. Some tests which may have influenced water movement in the soil were taken into consideration (hydraulic conductivity, infiltration and bulk density). Hydraulic conductivity can be defined as the ability of the fluid to pass through pores and fractured rocks. The infiltration rate measures how fast water enters the soil and the bulk density is the ratio of the mass to the bulk volume *V* of a given amount of soil [24]. Infiltration, hydraulic conductivity, and bulk density were done in three different land uses (forest, rangeland and agriculture) upstream (A), within (B) and downstream (C) (Figure 3) of the study area in order to compare how much land uses may influence the soil hydrological properties. The entire watershed (Kakia and Esamburmbur sub-catchments) had a unique soil texture at different depths, which was classified as clay loam (Table 3).

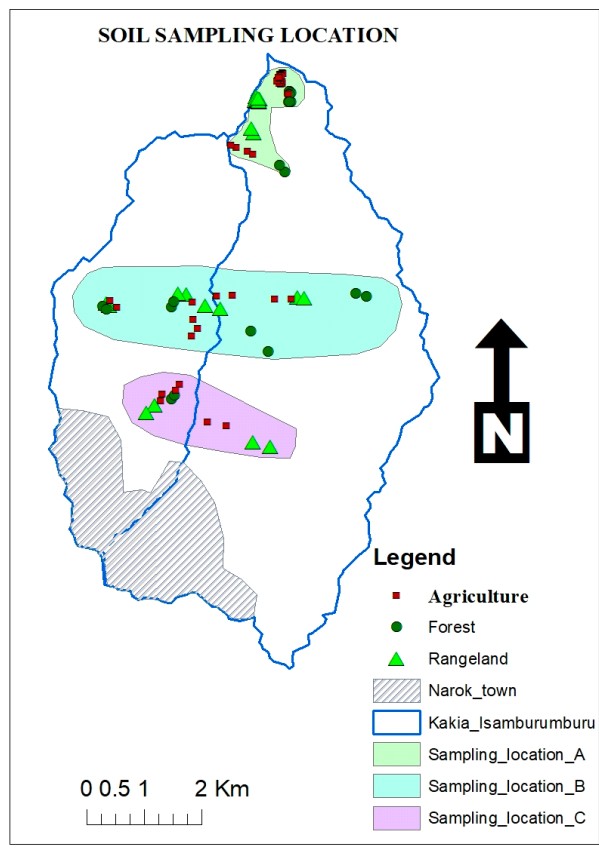

**Figure 3.** Soil sampling location.

**Table 3.** Site distribution according to land use (soil texture at all sites is clay loam).

|  | Upstream(A) | Within(B) | Downstream(C) |
|---|---|---|---|
| Forest | 2 sites | 4 sites | 1 site |
| Rangeland | 2 sites | 4 sites | 2 sites |
| Agriculture | 2 sites | 4 sites | 2 sites |

### 2.5.1. Infiltration

An infiltration test was done in different land uses in order to check how much it was affected by land use/cover. The infiltration rate was determined in the field using a double ring infiltrometer with falling water head method [25]. The rings were hammered into the soil at a depth of 15 cm and both were filled with water at an equal height. The water level was kept the same for both rings to allow the vertical infiltration from the inner ring. The drop of water level (cm) was measured and recorded after each 2 min of interval.

The infiltration rate was calculated by Equation (4):

$$I = D/T \text{ cm/hr} \tag{4}$$

where D = Depth of water that infiltrates the soil at certain time interval (cm) and T = time interval in hours.

### 2.5.2. Bulk Density

The bulk density is defined as the mass (weight) of a unit volume of dry soil [26]. Undisturbed soil was taken from three different depths (0–20 cm, 20–40 cm and 40–60 cm). The volume of the soil sample was determined by measuring the length (l) and diameter (2r) of the core (volume = $\pi r^2 l$) and the dry soil was weighed. Bulk density was calculated using Equation (5) [27]

$$\rho_b = \frac{M_s}{V_b} \tag{5}$$

$\rho_b$ = Soil bulk density, g/cm$^3$, $M_s$ = mass of dry soil, g, $V_b$ = volume of soil sample, cm$^3$

### 2.5.3. Hydraulic Conductivity

Undisturbed soil was taken from three different depths (0–20 cm, 20–40 cm and 40–60 cm) and a constant head method was used to measure the hydraulic conductivity of saturated soils in the laboratory based on the direct application of the Darcy equation [28] to a saturated soil column of a uniform cross sectional area.

A hydraulic head difference was imposed on the soil column and the resulting flux of water was measured. The conductivity is given by Equation (6):

$$K_{sat} = V.L/A.T.H \tag{6}$$

where *V* is the volume of water (Q) that flows through the sample of cross-sectional area *A* in time *T* and H is the hydraulic head difference imposed across a sample length *L*.

## 3. Results

### 3.1. Land Use Land Cover Classification

The land use maps developed for 1985, 1995, 2000, 2010 and 2019 and the land use change analysis are presented in Figures 3 and 4.

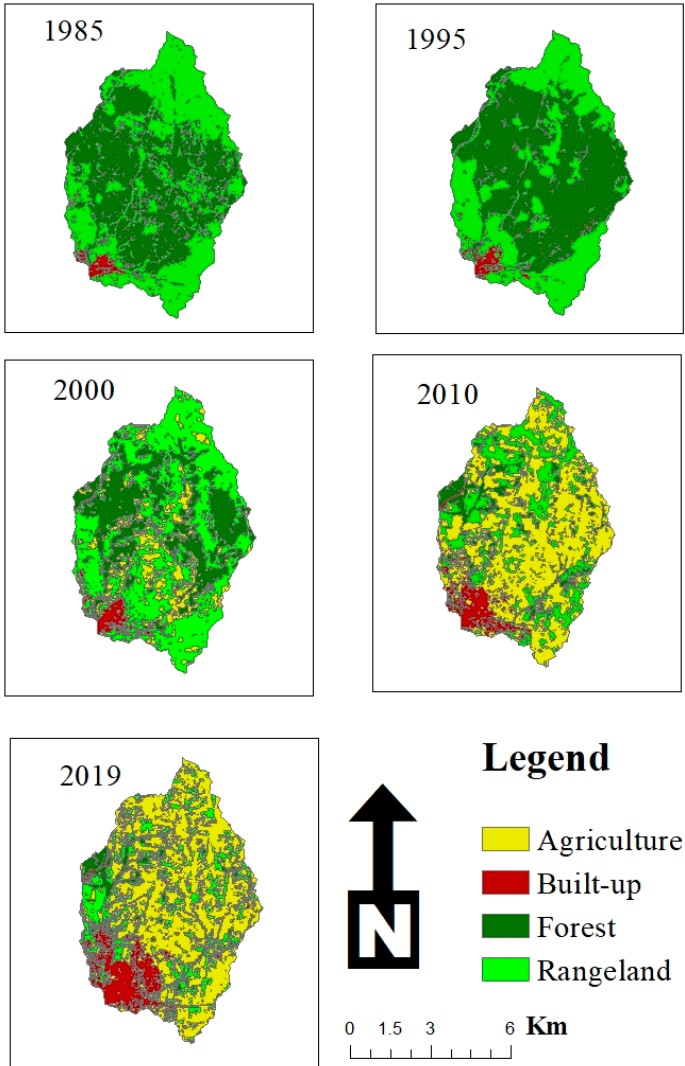

**Figure 4.** Classified land use land cover maps of Kakia and Esamburmbur sub-catchments of Enkale Narok watershed for 1985, 1995, 2000, 2010, 2019.

To understand the accuracy of land use land cover classification for the 1985, 1995, 2000, and 2010 classified images, producer's accuracy, user's accuracy and kappa, were summarized and are shown in Table 4 and the accuracy measures for the current classified image (2019) is summarized in Table 5. The overall classification accuracy of each Landsat image from 1985 up to 2019 are, respectively, 85%, 94%, 95%, 86%, and 90%. The respective kappa values are 0.71, 0.88, 0.91, 0.79, and 0.86. The results from the accuracy assessment were judged satisfactory and maps were then used for interpretation of land use changes in the watershed because all the classified images had an overall classification accuracy greater than or equal to 85% and a minimum Kappa of 0.71, which is classified as substantial [29].

**Table 4.** Accuracy assessment result of classified images.

|  | 1985 | | | 1995 | | | 2000 | | | 2010 | | |
|---|---|---|---|---|---|---|---|---|---|---|---|---|
|  | Pa | Ua (%) | (%) Ka | Pa | Ua (%) | (%) Ka | Pa | Ua (%) | (%) Ka | Pa | Ua (%) | Ka |
| Forest | 88.6 | 81.3 | 0.68 | 94.7 | 96.4 | 0.92 | 93.3 | 100 | 1.00 | 92.3 | 92.3 | 0.91 |
| Agriculture | - | - | - | - | - | - | 90.9 | 83.3 | 0.81 | 85.4 | 91.1 | 0.83 |
| Rangeland | 81.8 | 90.0 | 0.79 | 92.7 | 92.7 | 0.88 | 98.3 | 94.9 | 0.88 | 80.0 | 85.7 | 0.80 |
| Built-up | 100.0 | 100.0 | 1.00 | 100.0 | 66.7 | 0.66 | 50.0 | 100.0 | 1.00 | 100.0 | 64.3 | 0.61 |

**Table 5.** Accuracy assessment result of classified image (12 March 2019).

| Classe Names | Reference Totals | Classified Totals | No. Correct | Producers Accuracy | User Accuracy | Kappa |
|---|---|---|---|---|---|---|
| Forest | 11 | 11 | 11 | 100.0% | 100.0% | 1.00 |
| Agriculture | 20 | 18 | 17 | 85.0% | 94.4% | 0.91 |
| Rangeland | 11 | 13 | 9 | 81.8% | 75% | 0.68 |
| Built-up | 8 | 8 | 8 | 100.0% | 100.0% | 1.00 |
| Total | 50 | 50 | 45 | | | |

The percentages of each land use class are shown in Table 6. There was a remarkable change in land use/land cover in the study area (Figure 5), especially a decrease in forest and rangeland. From 1985 to 2019, the built-up area increased from 1.9% to 12.5% in the study area while agriculture increased from 0% to 55.2% in the study area during the same period. During the same period, forest and rangeland decreased from 46.6% to 6.2% and from 51.4% to 25.9% in the study area, respectively (Figure 6). As Figure 4 shows, there is hardly any natural forest remaining and what is available now is more of a thick shrubland. According to our interviews with local farmers, there were two group ranches, namely the Olopito group ranch and the Morijo group ranch. Agriculture started after the mid-1990s, after sub-division of land. As this study shows, in 2000, agriculture started increasing tremendously, leading to a remarkable decline in forest and rangeland (Figures 4 and 6).

**Table 6.** Percentage of land use/cover in Kakia and Esamburmbur sub-catchments of Enkale Narok watershed between 1985 and 2019.

| | 1985 | 1995 | 2000 | 2010 | 2019 | Change (1985–2019) |
|---|---|---|---|---|---|---|
| Forest | 46.5 | 54.5 | 28.3 | 9.6 | 6.2 | −40.3 |
| Rangeland | 51.6 | 43.1 | 53.1 | 28.4 | 25.9 | −25.7 |
| Agriculture | 0 | 0 | 15.6 | 54 | 55.2 | 55.2 |
| Built-up | 1.9 | 2.4 | 3.1 | 8 | 12.5 | 10.6 |

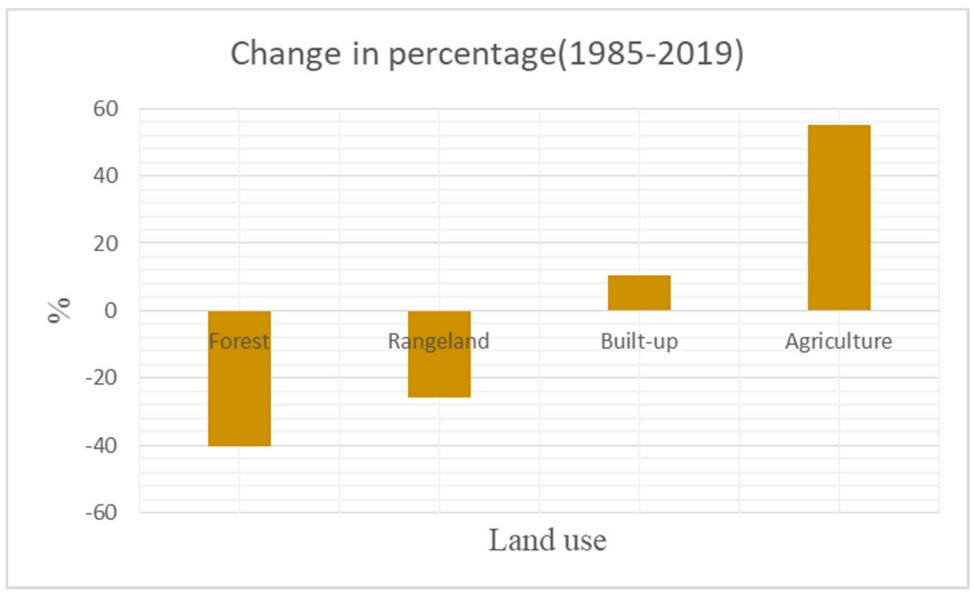

**Figure 5.** Percent land use/cover change (1985–2019) of Kakia and Esamburmbur sub-catchments of Enkale Narok watershed.

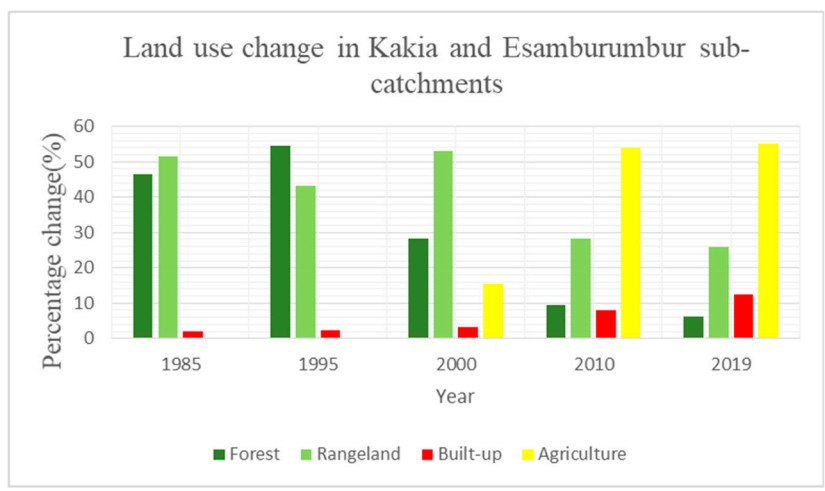

**Figure 6.** Land use land cover change in the study area.

### 3.2. Impact of Land Use/Land Cover Change on Soil Hydrological Properties

Our results show that the land use change in the study area may have had negative impacts on soil hydrological properties. We observed marked differences in soil hydrological properties for different land uses, which is an indicator of the impact of land use change on soil hydrological properties. Comparing the infiltration rates, hydraulic conductivity, and bulk density between the land uses, the basic infiltration rate was 89.14 cm/h in forest compared to 7.88 cm/h and 15 cm/h, respectively, in rangeland and agriculture (Table 7). The results of hydraulic conductivity and bulk density for various land uses are presented in Table 8. The hydraulic conductivity was found to be highest in forest than in rangeland and agriculture (Figure 7). For example, the average hydraulic conductivity in the forest was 46.29 cm/h at the top-soil layer (0–20 cm) compared to that of agriculture (4.58 cm/h) and rangelands (2.64 cm/h). For forest, the hydraulic conductivity decreased to 15.94 cm/h for the second depth (20–40 cm). At the same layer (0–20 cm), the bulk density was higher in rangeland (1.13 g/cm$^3$) and agriculture (1.06 g/cm$^3$) compared to the forest (0.72 g/cm$^3$).

**Table 7.** Basic infiltration for different land uses.

| | Basic Infiltration Rate (cm/h) | | |
|---|---|---|---|
| | **Agriculture** | **Rangeland** | **Forest** |
| Number of tests | 8 | 8 | 8 |
| Max | 30 | 12 | 156 |
| Min | 3 | 3 | 30 |
| Mean | 15 | 7.9 | 89.1 |
| Std. deviation | 7.9 | 3.3 | 48.4 |

**Table 8.** Characteristics of hydraulic conductivity and bulk density of Kakia and Esamburmbur sub-catchments.

| | Ksat(cm/h) | | | BD (g/cm³) | | |
|---|---|---|---|---|---|---|
| | 0–20 cm | 20–40 cm | 40–60 cm | 0–20 cm | 20–40 cm | 40–60 cm |
| **Agriculture** | | | | | | |
| Number of samples | 12 | 12 | 12 | 12 | 12 | 12 |
| Max | 11.5 | 7 | 5 | 1.21 | 1.21 | 1.19 |
| Min | 2 | 1.5 | 1.3 | 0.96 | 0.87 | 0.93 |
| Mean | 4.9 | 4.0 | 3.5 | 1.06 | 1.09 | 1.11 |
| Std Deviation | 3.0 | 1.9 | 1.1 | 0.07 | 0.1 | 0.09 |
| **Rangeland** | | | | | | |
| Number of samples | 9 | 9 | 9 | 9 | 9 | 9 |
| Max | 5.7 | 9.3 | 6.3 | 1.29 | 1.20 | 1.20 |
| Min | 0.4 | 0.8 | 0.7 | 0.84 | 0.86 | 0.93 |
| Mean | 2.6 | 4.4 | 3.7 | 1.13 | 1.07 | 1.08 |
| Std Deviation | 1.9 | 2.6 | 1.9 | 0.14 | 0.09 | 0.08 |
| **Forest** | | | | | | |
| Number of samples | 9 | 9 | 9 | 9 | 9 | 9 |
| Max | 86.4 | 35.5 | 19.2 | 0.91 | 1.12 | 1.14 |
| Min | 9.7 | 4.9 | 4 | 0.42 | 0.86 | 1 |
| Mean | 46.3 | 15.9 | 9.2 | 0.72 | 1.0 | 1.07 |
| Std Deviation | 21.6 | 9.7 | 5.2 | 0.14 | 0.10 | 0.06 |

Key: Ksat: Hydraulic conductivity, BD: Bulk density.

**Figure 7.** Comparison of hydraulic conductivity of the land uses at different depths of the catchment.

## 4. Discussion

The rapid deforestation in the study area from the 1990s and the sharp rise in agriculture can be attributed to the change in lifestyle of the local people (Maasai), who are predominantly pastoralists but who have recently ventured into commercial farming (mainly wheat) [30], as affirmed by elderly people in the watershed.

The observed increase in the built-up area may be partly attributed to an increase in income from mechanized agriculture in Narok County. Increased cash flow in Narok may have attracted people from other regions, thus causing a rise in population [31]. Population growth in Narok town may also be attributed to income generated and opportunities created at the Maasai Mara Game Reserve [32]. Narok is the gateway town to Maasai Mara Game Reserve and therefore, there are many business opportunities created by the thriving tourism industry. The rise in population created opportunities for business in the town, which consequently increased the demand for buildings for business premises and housing. Some of the businesses available in Narok include hotels, agro-vet shops, milk processors, tanneries, wheat and maize mills, bakeries, welding, motor garages, and carpentry [32]. Another cause of the decrease in forest is the trade of charcoal in Narok County, which has led to a rapid increase in the extraction of forest products [33].

Our results compare well with results from similar previous studies. For example, a study done by [34] on the effect of land use on infiltration in Taita Hills, Kenya, found that the bulk density was higher in grazing and cultivation lands. The mean bulk densities were 1.36 g/cm$^3$, 1.22 g/cm$^3$ and 0.85 g/cm$^3$, respectively, for grazing, cultivation and forest. The infiltration was higher in forest compared to that of other land uses. The mean basic infiltration rate values of forests, cultivations and grazing lands were 3926 mm/h, 1601 mm/h and 462 mm/h, respectively [34]. Forest has high infiltration rates, high hydraulic conductivity and low bulk density because there is minimal degradation, whereas permanent rangeland and agriculture have low infiltration and hydraulic conductivity and high bulk density because of a high compaction and loss of organic matter. Change in physical and chemical soil properties caused by land use land cover leads to soil degradation [7]. Compacted soil layers have less pore space and limit water movement through the soil profile [35]. A soil covered by vegetation increases infiltration and reduces runoff [36].

The infiltration rates in rangeland and agriculture were lower compared to the forest (Table 7). The low infiltration in these two land uses was caused by high compaction. Rangelands are used for grazing and the Maasai people are traditionally pastoralists and grazing is their primary livelihood. The number of livestock is an indicator of one's wealth. The grazing of livestock causes compaction of the soil which decrease the infiltration rates of the soil. Compaction in the agricultural land is caused by movement of agricultural machinery (i.e., tractors and mounted implements for ploughing, planting and weeding and harvesters). The results of bulk density of the soil (Figure 8) confirm that the reduction in soil infiltration in agricultural land and the rangeland can be attributed to compaction. There was a relatively low bulk density in forest compared to agriculture and rangeland (Figure 8). The compaction of the soil (by livestock or agricultural machinery) increases the bulk density of the soil, as observed in this study. The results obtained in this study compare well with other studies conducted elsewhere in the world. For example, a study carried out in Southern Spain on soil hydrological properties in different land uses showed that in intensive agriculture, the soil was more compacted (mean bulk density of 1.25 g/cm$^3$), while in forest, the soil had a lower bulk density (0.83 g/cm$^3$) [37]. Another study [38] done on the soil properties and water infiltration of Andosols in Tenerife (Canary Islands, Spain) found an increase in bulk density in cropped soil compared to the green forest. The infiltration rate was very rapid under green forest (796 mm h$^{-1}$) while in cropped soils, it was 67 mm/h [38].

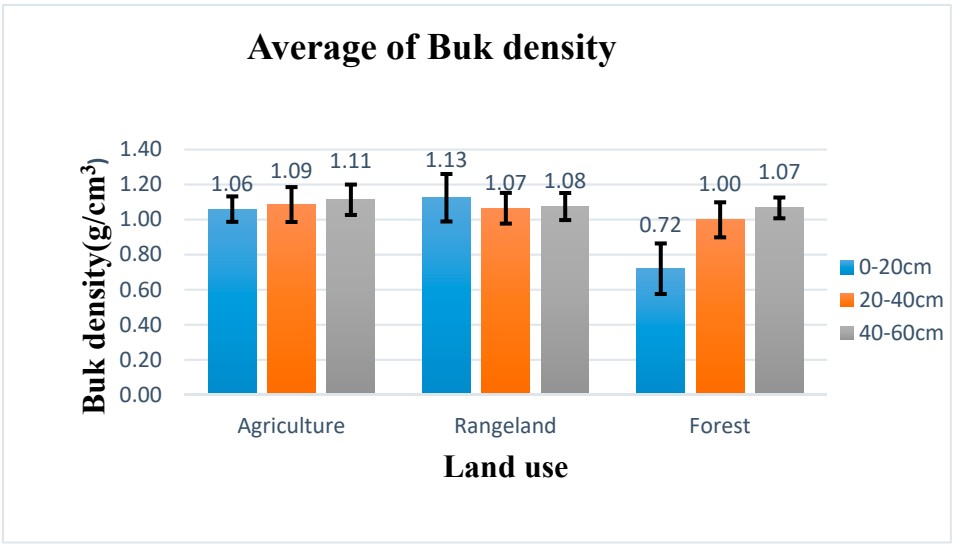

**Figure 8.** Comparison of bulk density of the land uses in different depths and different places of the catchment.

The results from a study conducted by [39] on the influence of land use and land cover on hydraulic and physical soil properties at the Cerrado Agricultural Frontier showed that the hydraulic conductivity in forest was 14.47 cm/h while in irrigated croplands, rainfed croplands and pasture, the hydraulic conductivity was 3.01 cm/h, 6.22 cm/h and 5.01 cm/h, respectively. A study done within the River Njoro watershed in Kenya showed similar results, where hydraulic conductivity was higher in forest (104 cm/h) than in grassland (69 cm/h) and in agriculture land (77 cm/h) [40].

Normally, the bulk density is expected to increase with depth, which was also generally observed in this study (Figure 8). For example, at 0–20 cm, the bulk density in forest was 0.72 g/cm$^3$, while at 20–40 cm and 40–60 cm, the bulk densities were 1 g/cm$^3$ and 1.07 g/cm$^3$ respectively (Figure 8). An opposite trend was observed, as expected, for hydraulic conductivity, which was found to generally decrease with depth (Figure 7). For example, in agriculture, the hydraulic conductivity was found to be 4.9 cm/h, 4.0 cm/h and 3.5 cm/h for 0–20, 20–40 and 40–60 depths respectively. For rangeland, the trends of variation of bulk density and hydraulic conductivity with depth were different (Figures 7 and 8). For rangeland, the bulk density was higher at the top layer (1.13 g/cm$^3$) compared to the next soil layer (1.07 g/cm$^3$). The hydraulic conductivity for the rangeland has a similar trend. The top layer had a lower hydraulic conductivity (2.6 cm/h) than the second layer (4.4 cm/h), confirming that the relatively high bulk density in the top layer was also manifested in the lower hydrologic conductivity for the top soil layer (Figures 7 and 8). The high bulk density at the top soil layer in rangeland may be attributed to compaction by livestock. The local community (Maasai) are typically pastoralists. The rangelands are permanently reserved for grazing livestock (mainly cattle and goats). The compaction by the hooves and the weight of this livestock increase the bulk density of the top layer of soil and seal the soil pores, which limits water movement. This was also observed in the infiltration experiments where the rangelands had the least infiltration compared to other land uses (Figure 9). In the rangeland, the infiltration curve was almost constant from the beginning (Figure 9), which can be explained by the compaction at the top-soil layer (Table 7) and the sealing of soil pores by the hooves of the animals. The drop of the infiltration curve from the initial infiltration rate to the basic infiltration rate was the lowest for rangelands compared to other land uses (Figure 9). This implies that compaction by the animals and sealing of soil pores by hooves on the top layer limited water movement in the soil (hydraulic conductivity) was the limiting factor for the downward movement of water in the soil (infiltration). The difference with compaction by agricultural machinery is that the tractors are heavy and the weight (compaction) is extended to deeper layers, unlike livestock (cattle and goats), that are relatively light and the compaction is only on the surface layer. The localised pressure from the hooves

cause more compaction and sealing of soil pores in the surface layer. This clearly shows the effect that stocking and overgrazing would have on soil properties, particularly the soil compaction and the limited infiltration rates. This limited infiltration rate would lead to a higher accumulation of runoff in the rangelands if a high (intensity) rainfall event occurs.

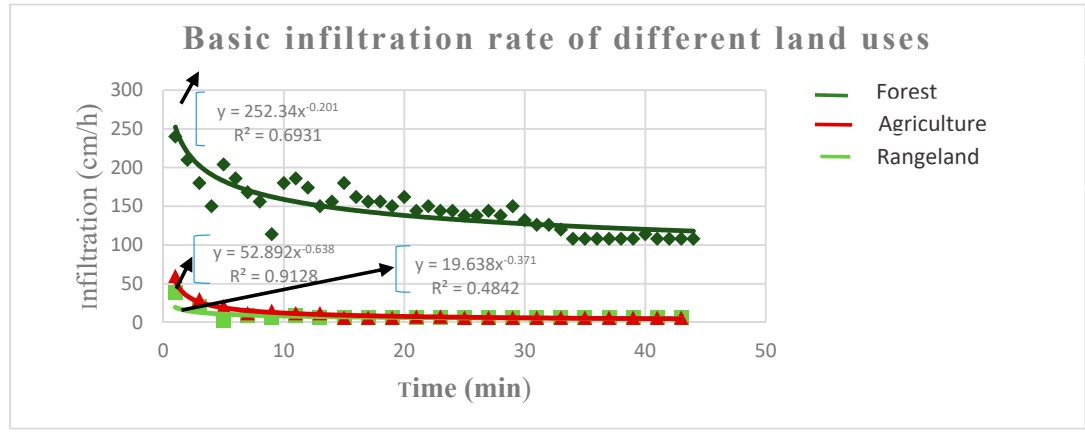

**Figure 9.** Comparison of basic infiltration rates of different land uses.

The results from this study show that land use has a major influence on soil hydrological properties (measured by water movement in the soil). Compaction of soil limits water movement, thus increasing the soil bulk density and decreasing hydraulic conductivity and infiltration. Forests had the least compaction and therefore, had high infiltration rates as well as high values of hydraulic conductivity compared to other land uses (agriculture and rangeland). This implies that land use changes have a high impact on soil hydrologic properties. Therefore, the observed rapid decrease in forest cover and increase in agricultural land in the watershed from 1985 to 2019 (Figure 6) had a highly negative effect on soil hydrological properties, which may have significantly contributed to the increase in floods in Narok town. It was also observed that Narok town increased significantly in size over the same period (1985–2019) and thus, the risk of flooding also increased. The flood risk has increased not only because of increased food hazard, but also due to the increase of vulnerability posed by a high population and a high number of buildings and property at risk of being destroyed by floods. The results of this study imply that any measures developed for flood management in Narok town should also include catchment conservation purposely targeted at reducing soil compaction, which would subsequently increase infiltration.

## 5. Conclusions

Supervised classification was used to classify and to report land use change over time. Between 1985 and 2019, 40.3% of the study area under forest was converted to other land uses, such as agriculture. Agriculture, which started in the late 1990s, currently occupies 55.2% of the study area. Rangeland decreased by 25.6% and built-up area increased by 10.6% in the study area. Soil tests (infiltration, hydraulic conductivity, and bulk density) in the various land uses present (forest, agriculture and rangeland) were carried out as indicators of how land use change affected the soil hydrological properties. Forest had the highest infiltration rate compared to both agriculture and rangeland. The mean basic infiltration in the study area was 89.1 cm/h in forest, while in rangeland and agriculture, it was 7.9 cm/h and 15 cm/h, respectively. For the top layer of the soil (0–20 cm) the rangeland had the highest bulk density (1.13 g/cm$^3$), followed by agriculture (1.06 g/cm$^3$), and forest had the lowest (0.72 g/cm$^3$). An opposite trend was observed for hydraulic conductivity, i.e., the values were 46.3 cm/h, 4.9 cm/h and 2.6 cm/h for forest agriculture and rangeland, respectively. This implies that the observed decrease in forest from 1985 to 2019 and the accompanying increase in agriculture negatively affected the soil hydrological properties. The low values of infiltration rates and hydraulic

conductivity in rangelands and agriculture as compared to that of forest are attributed to the compaction of soil by agricultural machinery and livestock, respectively. A peculiar trend on variation of bulk density and hydraulic conductivity with depth was found in rangeland. The top layer (0–20 cm) had the highest bulk density and lowest hydraulic conductivity compared to the next deeper layer (20–40 cm). This was attributed to the combined effect of compaction by livestock and sealing of soil pores by hooves of livestock. Unlike the case of tractor in agriculture, which is heavy and its weight (compaction) is transmitted into deeper soil layers, the cattle are lighter and the compaction is mainly limited to the top layer. The localized pressure impacted on the soil by the hooves contribute to sealing of soil pores. Degradation of the soil due land use change may have majorly contributed to the increase in the magnitude of runoff events in Narok town. The observed increase in built-up areas is mainly in Narok town, which was attributed to the increase in population and business opportunities in the town due to the increased income from large-scale wheat farming and tourism activities in the Maasai Mara Game Reserve. The results of this study show that there is ab increased flood hazard in Narok town due to the degradation of the catchment and increased vulnerability to floods due to the increase of built-up areas (show of increase of population and business activities), which essentially increases the flood risk in Narok town. Based on the findings of this study, flood management strategies for Narok town should include catchment management strategies that reduce soil compaction and increase soil infiltration.

**Author Contributions:** N.M.M., H.M.M., J.K.M. and J.M.G. conceptized the idea and designed the research. N.M.M. carried out the experiments, analysis and wrote the first draft manuscript. H.M.M., J.K.M. and J.M.G. edited the manuscript.

**Funding:** This research was funded by the Pan African University of Basic Sciences, Technology and Innovation (PAUSTI).

**Acknowledgments:** The authors acknowledge and appreciate the financial support by National Research Fund (Kenya) (Narok Flash Floods Management Project) and administrative support by Pan African University of Basic Sciences, Technology and Innovation (PAUSTI).

**Conflicts of Interest:** The authors declare no conflicts of interest.

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
