# Peer review of "Analysis of Land Use Change and Its Impact on the Hydrology of Kakia and Esamburmbur Sub-Watersheds of Narok County, Kenya"

_hydrology, doi:10.3390/hydrology6040086_

Round 1

Reviewer 1 Report

I assess the paper „Analysis of land use, land cover change and its impact on the hydrology of Kakia and Esamburumbur sub-watersheds of  Narok County, Kenya”  as original and important because of the importance of the problem it raises. It describes the environmental effects of changes in land use. The article focuses on soil properties. It points to the enormous importance of forest areas for protection against flood risk. The article considers also environmental changes against the background of social changes.

The role of the reviewer is to indicate the discussion elements of the study. I must point out that my point of view is the point of view of a hydrogeologist.

I have some doubts about the units of hydraulic conductivity in lines 76-78: cm3/h. In hydrogeology the unit of saturated hydraulic conductivity is length/time. Although the author refers to the work of Ceyhun Gol, in the same work in Table 1 the unit of Ksat is already correct (cm / h).

The analysis of land-use changes in the years 1985 - 2019 is detailed and correct.

I suggest using "dry bulk density" or "density of dry soil" instead of bulk density - in geology bulk density is the mass of moist soil up to its volume, e.g. for sands the value of bulk density is 1.65-2.00 g/cm3 depending on soil moisture and soil compacting.

The authors examined hydraulic conductivity, infiltration and dry bulk density for the purposes of the study. These parameters depend on the granulometric composition of the soil. Meanwhile, the authors assume that they depend only on land use. In my opinion, the key factor here is the type of soil. The infiltration will be less on clay soils than on sandy soils. Therefore, I suggest that the interpretation of the results take into account the type of soil (grain-size distribution) of the layers at a depth of 0-20, 20-40, 40-60 cm.

In my opinion, the authors should document the results of soil parameters measurements more accurately. I propose to attach a table with the results of measurements, soil type in individual layers, and information on land use. This table may contain synthetic data, where the starting point will be soil types (e.g. sandy soils), range of: Ksat, dry BD, infiltration (min, max, average, st. deviation) for each type of land use. It would be ideal to show the hydrogeological parameters for the same type of soil used in different ways. Tables 6 and 7 should provide information on the number of samples.  I propose to provide maximum and minimum values in Table 7.

Please draw points for each curve in Figure 9 (R2 = 0.48 is not high). I suggest changing the color of the curve for rangeland. The location of the equations should clearly refer to the curves.

As I have mentioned, my suggestions will strengthen the hydrological aspect of this study.

I would consider shortening the title of the paper: "Analysis of changes in land use and their impact on the hydrology of Kakia and Esamburumbur sub-watersheds of Narok County, Kenya”
Land-use change usually results in a change in land cover. 

Other technical notes:

Line 107: the reference to literature should be unified (Okayo et al., 2015).

Lines 181-183: explanations should be given in equations No: 1, 2, 3.

Line 157: The Maximum Likelihood algorithm should have a reference to the literature.

There is no reference to Figure 2 in the text.

Authors should use a linear map scale in figure 4.

Areas A, B, C marked in Figure 3 should be described in the explanations.

Line 413: no Figure number in reference.

Line 233: literature reference [39] cited as Landon (1984).

I have often found a lack of spaces when referring to the literature.

Author Response

We think the reviewer for reading our work and we appreciate that he/she found our work original and important. Find attached our response to the comments raised by the reviewer.

Reviewer 2 Report

The article entitled” ANALYSIS OF LAND USE LAND COVER CHANGE AND ITS IMPACT ON THE HYDROLOGY OF KAKIA AND ESAMBURUMBUR SUB-WATERSHEDS OF NAROK COUNTY, KENYA”

Is generally well written. This study was to investigate how land use/land cover has changed in Kakia and Esamburumbur sub-catchments in Narok during the past three decades and their relation with increasing flooding . The authors provided detailed information in the introduction section with convincing discussions on previous studies and thus the objective was reasonable and rendered clearly. The methods were also clearly described and understandable. Results and discussions were connected closely. However, the main problem is the novelty of the study. It is recommended that the authors state what is new compared with previous studies in your study. For example, are your research methods different from others?  Another recommendation is to analysis the data by the two different sub-catchments as well as by upstream, within and downstream. For example, are there significant differences of mean hydraulic properties between the two sub-catchments or between A, B and C?

There are several minor insures:

Fig. 1 Please provide a higher quality fig. Fig. 3 Please add name for A B and C. Fig. 7 and 8 Please added error bars.

Author Response

We thank the reviewer for hearing from him/her that the information in the introduction section is detailed with convincing discussions on previous studies and the objective is reasonable and rendered clearly. Find attached our response to the comments raised by the reviewer.

Round 2

Reviewer 2 Report

The manuscript was improved to some extent. However, the error bar added is not correct. It is simple to add an error bar in excel or other software. Please search online to see what error bar looks like.
